# Multi-View Convolutional Neural Networks in Rupture Risk Assessment of Small, Unruptured Intracranial Aneurysms

**DOI:** 10.3390/jpm11040239

**Published:** 2021-03-24

**Authors:** Jun Hyong Ahn, Heung Cheol Kim, Jong Kook Rhim, Jeong Jin Park, Dick Sigmund, Min Chan Park, Jae Hoon Jeong, Jin Pyeong Jeon

**Affiliations:** 1Department of Neurosurgery, College of Medicine, Hallym University, Chuncheon 24252, Korea; sparkahn@naver.com; 2Department of Radioilogy, College of Medicine, Hallym University, Chuncheon 24252, Korea; khc@hallym.or.kr; 3Department of Neurosurgery, College of Medicine, Jeju National University, Jeju 63243, Korea; pedineur@daum.net; 4Department of Neurology, Konkuk University Medical Center, Seoul 05030, Korea; medicalstory@gmail.com; 5AIDOT Inc., Seoul 05854, Korea; dsigmund@aidot.ai (D.S.); parkminchan@aidot.ai (M.C.P.); jman@aidot.ai (J.H.J.); 6Genetic and Research Inc., Chuncheon 24252, Korea

**Keywords:** intracranial aneurysm, convolutional neural network, angiography

## Abstract

Auto-detection of cerebral aneurysms via convolutional neural network (CNN) is being increasingly reported. However, few studies to date have accurately predicted the risk, but not the diagnosis itself. We developed a multi-view CNN for the prediction of rupture risk involving small unruptured intracranial aneurysms (UIAs) based on three-dimensional (3D) digital subtraction angiography (DSA). The performance of a multi-view CNN-ResNet50 in accurately predicting the rupture risk (high vs. non-high) of UIAs in the anterior circulation measuring less than 7 mm in size was compared with various CNN architectures (AlexNet and VGG16), with similar type but different layers (ResNet101 and ResNet152), and single image-based CNN (single-view ResNet50). The sensitivity, specificity, and overall accuracy of risk prediction were estimated and compared according to CNN architecture. The study included 364 UIAs in training and 93 in test datasets. A multi-view CNN-ResNet50 exhibited a sensitivity of 81.82 (66.76–91.29)%, a specificity of 81.63 (67.50–90.76)%, and an overall accuracy of 81.72 (66.98–90.92)% for risk prediction. AlexNet, VGG16, ResNet101, ResNet152, and single-view CNN-ResNet50 showed similar specificity. However, the sensitivity and overall accuracy were decreased (AlexNet, 63.64% and 76.34%; VGG16, 68.18% and 74.19%; ResNet101, 68.18% and 73.12%; ResNet152, 54.55% and 72.04%; and single-view CNN-ResNet50, 50.00% and 64.52%) compared with multi-view CNN-ResNet50. Regarding F1 score, it was the highest in multi-view CNN-ResNet50 (80.90 (67.29–91.81)%). Our study suggests that multi-view CNN-ResNet50 may be feasible to assess the rupture risk in small-sized UIAs.

## 1. Introduction

Intracranial aneurysms (IAs) have been automatically detected using various algorithms based on magnetic resonance angiography (MRA) or computed tomography angiography (CTA). Previous works in the field have been successful at diagnosing and detecting aneurysms [1,2]. It is important to decide whether or not to treat unruptured intracranial aneurysms (UIAs) as well as automatically detect aneurysms clinically. Although UIAs are associated with a relatively lower rupture rate of less than 2%, ruptured aneurysms increase the mortality rate above 50% within the first six months, and morbidity is a major issue [3,4,5]. Moreover, higher rates of adverse effects (10%) after the treatment [6,7] also interfere with the treatment of UIAs, in particular, asymptomatic aneurysms before rupture. Therefore, it may be more helpful to select UIAs indicated for treatment based on risk classification using machine learning algorithms rather than simple automatic diagnosis [8].

Previously, we introduced single-view CNN for diagnosing aneurysm rupture (vs. UIA) based on three-dimensional (3D) digital subtraction angiography (DSA), which had a sensitivity of 78.76%, a specificity of 72.15%, and an overall diagnostic accuracy of 76.84% [5]. However, risk prediction based on a single input image may have limited value in clinical settings because physicians acquire multiple 3D-DSA images in multiple directions to determine the treatment plan, instead of single images. Moreover, in the previous study, there was no study that evaluated the risk of rupture in only UIAs. Multi-view CNN allows clinicians to draw conclusions based on various scanned images in multiple directions. Accordingly, compared with single-view CNN, multi-view CNN may be more useful in treatment decision-making for UIAs. Here, we propose a multi-view CNN in an effort to predict the rupture risk (high vs. non-high) of small-sized UIAs in the anterior circulation using 3D-DSA images.

## 2. Materials and Methods

### 2.1. Datasets

In this study, 3D-DSA images were used to develop the CNN algorithm. The images were acquired consecutively at three hospitals. Two datasets were prepared from the acquired images of training and test datasets. The training dataset included UIAs from January 2012 to December 2017, and the test dataset included those acquired from January 2018 to December 2019. Inclusion criteria were (1) adult patients more than 18 years old; (2) saccular aneurysm; (3) anterior circulation aneurysm; and (4) small aneurysms less than 7 mm in maximal diameter [5]. We excluded (1) fusiform and dissecting aneurysms; (2) traumatic aneurysms; (3) posterior circulation aneurysms; and (4) treated aneurysms by clipping or coiling.

A total of 457 UIAs consisting of 364 UIAs of the training dataset and 93 UIAs of the test dataset were included. Baseline characteristics of the two datasets are presented in Table 1. The size of the aneurysms was 5.2 ± 1.2 mm in the training dataset and 5.3 ± 1.3 mm in the test dataset. The UIAs in the training dataset were located in anterior cerebral artery (ACA) (*n* = 78, 21.4%), middle cerebral artery (MCA) (*n* = 115, 31.6%), and internal carotid artery (ICA) (*n* = 171, 47.0%). The distribution of the UIAs in the test dataset is as follows: ACA, *n* = 15 (16.1%); MCA, *n* = 40 (43.0%); and ICA, *n* = 38 (40.9%). The number of high-risk UIAs included 133 (36.5%) in the training and 44 (47.3%) in the test datasets. A total of 251 and 206 images were obtained from Siemens Healthcare and Philips Medical System, respectively.

We developed an automatic risk classification system for small-sized UIAs based on representative images with six directions acquired with 3D-DSA [5]. High-risk aneurysms were defined as (1) those with an irregular aneurysm wall with small bleb(s); (2) secondary aneurysms protruding from the fundus of saccular aneurysm; (3) bi- or multilobular aneurysms; and (4) those with an aspect ratio ≥ 1.6 [9,10] (Appendix A). Risk interpretation of the UIAs was conducted by two blinded readers (KHC and JPG). Disagreements were resolved by the third reviewer (JKL) (Appendix A). Image acquisition and post-image processing are presented in the Appendix A. Briefly, DSA procedures were conducted with the Axiom Artis Zee (Siemens Healthcare, Erlangen, Germany) or the Allura Xper FD 20/20 (Philips Medical Systems, Best, The Netherlands) with standard injection protocols as described previously [5]. Post-processing of the 3D-DSA was conducted in an independent workstation equipped with InSpace 3D software [11,12,13]. The study design was listed on Clinical Research Information Service (registration number KCT0005084) prior to initiation of the study.

### 2.2. Multi-View Convolutional Neural Networks

We used a multi-view CNN architecture to assess the rupture risk in small-sized UIAs. Regions of interest (ROIs) including the aneurysms were selected by the neurosurgeon or neuroradiologist, extracted, and the cropped ROI was entered in parallel as input for the six neural networks, followed by convolution and pooling of the layers in each neural network. The multi-view CNN received six three-channel 224 ××× 224-sized input images, extracted from each different ROI view [5]. In the end, the pooled view was used to merge all six neural networks into a single one (Figure 1 and Appendix A). More specifically, each view model in the multi-view CNN was initialized with the weights pre-trained from single-view CNN after feature extraction. During the multi-view CNN training, each view model weight was frozen, whereas the pooled layer view was fine-tuned. Hyper-parameters are combinations of values that are defined for training on CNN. The learning rate scheduler was applied. The learning rate varies depending on the setting of the learning rate scheduler. The learning rate warm-up was 5 epochs, and cosine decay was introduced after reaching the maximum learning rate. The total epoch was set to 100, and the cross-entropy function was selected as the loss function. Adam was used for the optimizer [14]. Additionally, all weights were initialized with ImageNet weights [15]. The batch size was set as 32. All training in the multi-view CNN was conducted with each aneurysm and not each image. Finally, label smoothing was added for the training [16].

ResNet50, AlexNet, and VGG16 were used as the components of multi-view CNN architecture for risk classification. ResNet101 and ResNet152 were then compared with ResNet50. Additionally, the diagnostic performance of multi-view CNN-ResNet50 was compared with that of single-view CNN-ResNet50. Pytorch was used as the main training framework [17].

### 2.3. Statistical Analysis

Descriptive analysis is presented as the numbers of subjects (percentage) for discrete and categorical variables and mean with standard deviation (SD). Two-by-two tables were generated to assess sensitivity, specificity, and overall accuracy [11]. The degree of agreement between the two readers was assessed using the k test (Appendix A) [14]. Confidence interval (CI) estimation was performed using the binomial proportions confidence interval, specifically the Clopper–Pearson interval, in which the success probability was estimated by the total number of trials and the number of successful trials based on the cumulative probabilities of the binomial distribution. A *p*-value < 0.05 was considered statistically significant. The analysis was conducted using SPSS version 19 (IBM, Armonk, NY, USA) and MedCalc (www.Medcalc.org, accessed on 1 February 2021).

## 3. Results

### 3.1. ResNet50 vs. AlexNet vs. VGG16

Comparative analyses of three different multi-CNN models of ResNet50, AlexNet, and VGG16 were performed. During the training, we achieved over 99% accuracy (Appendix A). For the test, we evaluated every single model from each epoch, and each model in 93 UIAs to identify the best model (Table 2). Multi-view CNN-ResNet50 demonstrated a sensitivity of 81.82 (66.76–91.29)%, a specificity of 81.63 (67.50–90.76)%, and an overall accuracy of 81.72 (66.98–90.92)% (Table 3 and Figure 2). ResNet50 exhibited superior sensitivity in terms of risk prediction compared with other types of CNNs such as AlexNet (63.64 (47.74–77.17)%) and VGG16 (68.18 (52.29–80.93)%), but similar specificity. In terms of overall accuracy, ResNet50 was better than AlexNet (76.34 (62.31–88.19)%) and VGG16 (74.19 (58.93–85.60)%).

### 3.2. ResNet50 vs. ResNet101 vs. ResNet152

We compared the diagnostic performance of ResNet50 with the same type but different layer of ResNet101 and ResNet152. Detailed configurations of the three ResNets are presented in Table 4. ResNet50 was superior to ResNet101 (68.18 (52.29–80.93)%) and ResNet152 (54.55 (39.00–69.31)%) after the 50th epoch. ResNet50 had higher overall accuracy than ResNet101 (73.12 (57.71–84.66)%)) and ResNet152 (72.04 (58.18–85.68)%) (Table 3 and Figure 2). ResNet50 (80.90 (67.29–91.81)%) also exhibited higher F1 score than ResNet101 (70.59 (55.42–84.28)%) and ResNet152 (64.86 (47.46–79.79)%).

### 3.3. Multi-View CNN-ResNet50 vs. Single-View CNN-ResNet50

We further compared the diagnostic performance of multi-view CNN-ResNet50 and single-view CNN-ResNet50. Single-view CNN-ResNet50 showed a sensitivity of 50.00 (34.79–65.21)%, a specificity of 77.55 (63.01–87.75)%, and an overall accuracy of 64.52 (48.93–78.45)%, which were lower than those of multi-view CNN-ResNet50.

## 4. Discussion

To the best of our knowledge, this is the first study using multi-view CNN for the prediction of UIA risk. Multi-view CNN-ResNet50 demonstrated a sensitivity of 81.82%, a specificity of 81.63%, overall accuracy of 81.72%, and an F1 score of 80.90% for the classification of rupture risk involving small-sized UIAs.

Various machine learning algorithms have been increasingly implemented in disease diagnosis and predicting disease risk. Previous machine learning algorithms mainly focused on the auto-detection of cerebral aneurysms, but not risk prediction [1,18,19,20,21,22,23]. Based on MIP images acquired via MRA, CNN yielded superior detection rates (>90%) of aneurysm [1,19]. Recently, a two-stage CNN consisting of region localization and aneurysm detection was introduced for the auto-detection of aneurysms [24]. The regional average grayscale suppression was used to differentiate ROI of aneurysm and enlargement area [24]. Hu et al. [25] used Bayesian optimization for aneurysm detection based on DSA images, showing a sensitivity of 96.4% with a false-positive rate of 6.2%. However, in most cases in daily practice, DSA is additionally performed to acquire detailed information of aneurysm including relationship with nearby arteries to decide the treatment plan [26], but not the diagnosis of aneurysm itself [5]. Therefore, CNN based on DSA images can be used more frequently in clinical practice for the prediction of UIA risk, but not auto-detection. Kim et al. [5] proposed single-view CNN for differentiating ruptured aneurysm from UIA using DSA images based on AlexNet_v2 architecture. A single input image after data preprocessing using a histogram-oriented gradient resulted in two output parameters of ruptured aneurysm or UIA. Single-view CNN yielded better diagnostic accuracy than human evaluators based on the AUROC difference of 0.163 (*p* < 0.001) [5]. Nonetheless, its clinical usefulness is limited because the rupture risk of an aneurysm is evaluated by the clinician based on multiple images acquired simultaneously, and not each individual image. Therefore, we built six independent neural networks for six views in parallel and were trained separately in the front layers to evaluate the advantage of multi-view CNN for prediction of the risk of aneurysm rupture in patients with small-sized UIAs. Subsequently, the six individual models were combined together by pooling the views in the last layers.

Since 2012, most CNN algorithms have been evaluated using the ImageNet Large Scale Visual Recognition Challenge (ILSVRC). AlexNet, VGG16, and ResNet50 were regarded as important CNN architectures and milestones (Table 4). As the 2012 ILSVRC winner, AlexNet represents a base architecture for various CNN models [27]. VGG16, which is ranked 2nd in 2014 ILSVRC, uses a different approach, with a reduced filter size [28]. Compared with AlexNet’s parallel architectures, VGG16 adopted a single architecture with smaller 3 × 3 filters, which reduced the computational cost and time. ResNet, ranked 1st in the 2015 ILSVRC, provided a completely different solution to the classification task, and suggested a residual block, with superior performance compared with others [29,30,31]. Compared with the plane layers, ResNet’s residual block included the identity block that preserves the existing information. As a result, both existing and newly trained information contributed to the total performance. ResNet can be differentiated into ResNet50, ResNet101, and ResNet152 depending on the number of layers. Additional layers require more computation and time. In many cases, image resolution is enhanced with architectures comprising a large number of layers, e.g., ResNet152. However, small architectures such as ResNet50 yielded better results involving low-resolution images. In our study, the average resolution of the cropped aneurysm was quite low, and therefore, ResNet50 predicted the rupture risk more accurately than ResNet101 and ResNet152 in small-sized UIAs. The superiority of ResNet50 compared with ResNet101 and ResNet152 can also be attributed to fewer trainable parameters in ResNet50. In our fine-grained classification task, ResNet50 with 23 million parameters could be optimized more effectively than ResNet152 with 60 million parameters.

The edges in 3D dimensions and roughness of the aneurysm surface are indicators for the assessment of rupture risk as well as aneurysm shape. As a main algorithm, we acquired six independent neural networks for six views in parallel, and they were trained separately in the front layers and combined via view pooling in the final layers. Clearer features were extracted in the front layers using this method, and the relationships between views were trained in the end as well. In short, both similar feature extractions and the relationship between the views enhanced accurate prediction of rupture risk using our multi-view CNN. Accordingly, multi-view CNN can be fine-tuned using pre-trained weights resulting in higher precision of risk prediction involving small-sized UIAs compared with single-view CNN. Kang et al. [32] also demonstrated higher accuracy of lung nodule classification using 3D multi-view CNN compared with single-view CNN. In addition, it is possible to apply our CNN system for rupture risk regardless of angiography machine due to post-processing of images via histogram equalization. Therefore, multi-view CNN will be more useful for clinicians who are less experienced in deciding the treatment plan for patients with UIAs.

In this study, we only included anterior circulation aneurysms measuring less than 7 mm in size. Tykocki et al. [33] reported that morphological factors such as parent artery size and aspect ratio varied between anterior and posterior circulation aneurysms. Usually, UIAs greater than 7 mm are indicated for treatment due to the risk of rupture [34,35]. However, the treatment policy for an aneurysm less than 7 mm in size is disputed. According to the second international study of unruptured intracranial aneurysms (ISUIA), small aneurysms measuring less than 7 mm without previous subarachnoid hemorrhage (SAH) are associated with 0% cumulative rupture rate over a period of 5 years [36,37]. Nevertheless, a small UIA can rupture during the follow-up period. Suzuki et al. [38] reported that bifurcated aneurysms and UIAs harboring blebs were associated with future rupture. Therefore, it may be helpful to selectively treat UIAs with a high risk of rupture in the future via CNN. Compared with other machine learning techniques, CNN is substantially more amenable to medical image processing and classification tasks, as well as easy customization for specific tasks. Our study demonstrated that ResNet50-based multi-view CNN had higher accuracy than AlexNet, VGG16, ResNet101, ResNet152, or single-view CNN.

The study limitations are as follows. First, the study included only small-sized UIAs in the anterior circulation. Morphometric variables, such as aspect ratio and parent artery size, which are associated with aneurysm rupture, vary according to aneurysm location and size [5]. Accordingly, the evaluation of an algorithm for rupture risk classification involving posterior circulation aneurysm is further required. Second, images acquired under specific conditions such as concomitant vascular disease or cerebral angiograms from General Electric Healthcare were not included. Thus, there may be some limitations in terms of real-world clinical effectiveness of our multi-view CNN. Third, no quantitative hemodynamic parameters were considered. In addition to morphological characteristics, hemodynamic factors such as wall share stress (WSS), oscillatory shear index, WSS gradient, or spatial WSS gradient are associated with aneurysm rupture. Although Chen et al. [39] did not report that machine learning was better than conventional logistic regression in predicting UIA rupture, a model is needed based on morphological and hemodynamic features in subsequent studies using artificial intelligence approaches. Fourth, we used histogram equalization rather than vesselness filtering. The main purpose of the histogram equalization applied in this study was to normalize the images obtained from two different angiography machines. In addition, we aimed to identify the risk of aneurysm rupture via the pattern and shape of the aneurysm, not the vessel itself. In future work, we will continue to explore other filtering methods to normalize images from different types of angiography machines as well as vesselness filtering to increase the diagnostic accuracy for high-risk aneurysms. Finally, we applied a user-selected ROI rather than automated diagnosis around the UIAs in DSA images for risk prediction. Clinically, DSA is generally used to decide treatment strategies rather than investigate the diagnosis [5]. Nonetheless, a study analyzing risk prediction of UIA based on automated detection is necessary to improve the clinical effectiveness of the CNN system.

## 5. Conclusions

We proposed a new multi-view CNN to evaluate rupture risk in small UIAs based on 3D-DSA images. The diagnostic performance of multi-view ResNet50 was better than that of the other types of CNNs. An external study is required to further corroborate our results before practical, clinical, and commercial applications can be envisaged.

## Figures and Tables

**Figure 1 jpm-11-00239-f001:**
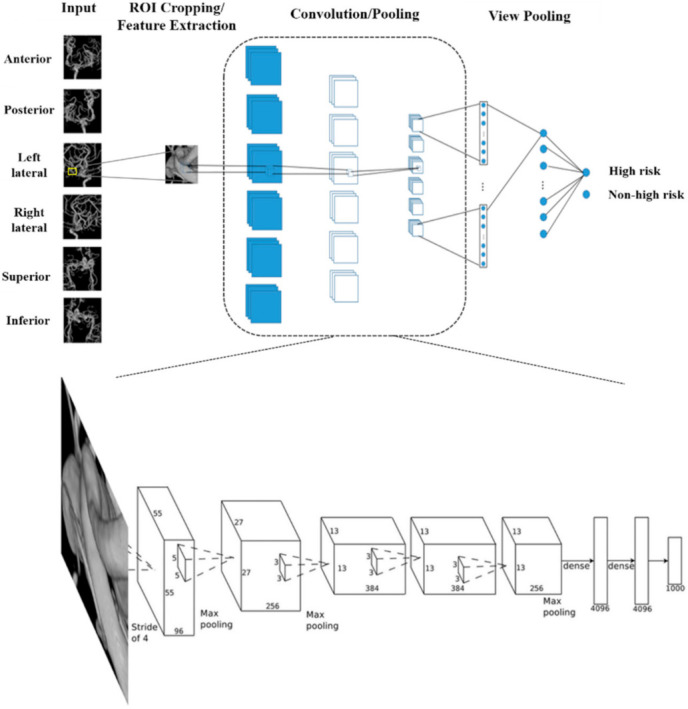
Overview of our proposed multi-view CNN architecture and simplified example of the convolution and pooling layers.

**Figure 2 jpm-11-00239-f002:**
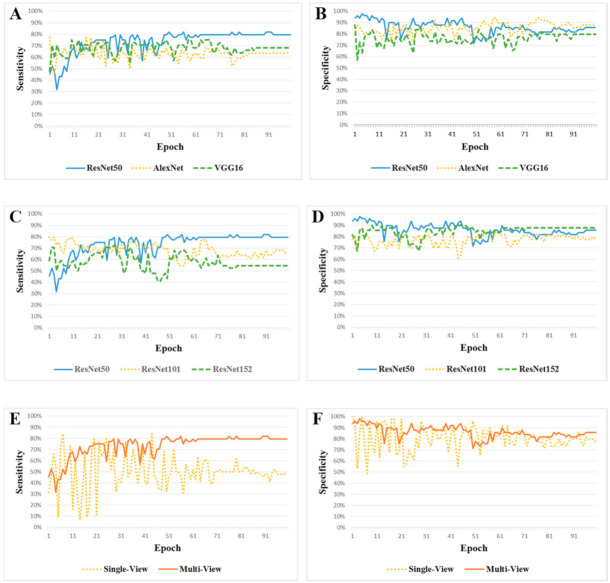
Comparative analyses of multi-view CNN-ResNet50 for predicting rupture risk type involving small UIAs under various conditions: other types of AlexNet and VGG16 (**A**,**B**), same type but different layers (ResNet101 and ResNet152) (**C**,**D**), and single-view CNN (**E**,**F**).

**Table 1 jpm-11-00239-t001:** Clinical and radiological characteristics: training vs. test datasets.

Variables	Training Dataset (*n* = 364)	Test Dataset (*n* = 93)
Clinical findings		
Female	206 (56.6%)	54 (58.1%)
Age, years	57.2 ± 14.5	56.8 ± 15.3
Hypertension	97 (26.7%)	28 (30.1%)
Diabetes mellitus	35 (9.6%)	10 (10.8%)
Hyperlipidemia	42 (11.5%)	10 (10.8%)
Smoking	48 (13.2%)	12 (12.9%)
Radiologic findings		
High-risk UIA	133 (36.5%)	44 (47.3%)
Size (mm)	5.2 ± 1.2	5.3 ± 1.3
Aneurysm location		
Anterior cerebral artery	78 (21.4%)	15 (16.1%)
Middle cerebral artery	115 (31.6%)	40 (43.0%)
Internal carotid artery	171 (47.0%)	38 (40.9%)
Imaging platform		
Siemens Healthcare	210 (57.7%)	41 (44.1%)
Philips Medical Systems	154 (42.3%)	52 (55.9%)

**Table 2 jpm-11-00239-t002:** Accuracy of multi-view CNN-ResNet50 for the prediction of rupture.

	CNN	ResNet50 Classification		
Diagnosis		High Risk	Non-High Risk	Total
High risk		36	8	44
Non-high risk		9	40	49
Total		45	48	93

**Table 3 jpm-11-00239-t003:** Binary classification of rupture risk involving small UIAs in the test dataset using various CNN architectures. CI, confidence interval.

Type	Models	Sensitivity (95% CI)	Specificity (95% CI)	Overall Accuracy (95% CI)	F1 Score (95% CI)
Multi-view	ResNet50	81.82 (66.76–91.2)%	81.63 (67.50–90.76)%	81.72 (66.98–90.92)%	80.90 (67.29–91.81)%
Multi-view	AlexNet	63.64 (47.74–77.17)%	87.76 (74.54–94.92)%	76.34 (62.31–88.19)%	71.79 (55.13–85.00)%
Multi-view	VGG16	68.18 (52.29–80.93)%	79.59 (65.24–89.28)%	74.19 (58.93–85.60)%	71.43 (55.42–84.28)%
Multi-view	ResNet101	68.18 (52.29–80.93)%	77.55 (63.01–87.75)%	73.12 (57.71–84.66)%	70.59 (55.42–84.28)%
Multi-view	ResNet152	54.55 (39.00–69.31)%	87.76 (74.54–94.92)%	72.04 (58.18–85.68)%	64.86 (47.46–79.79)%
Single-view	ResNet50	50.00 (34.79–65.21)%	77.55 (63.01–87.75)%	64.52 (48.93–78.45)%	57.14 (40.82–73.69)%

**Table 4 jpm-11-00239-t004:** Detailed configuration of ResNet50, ResNet101, ResNet152, AlexNet, and VGG16. Convolution and pooling layers of ResNet block 2, 3, 4, and 5 were performed with a stride of 2.

Block	Output Size	ResNet50	ResNet101	ResNet152	VGG16	Output Size	AlexNet
1	112 × 112	7 × 7, 64, stride 2	7 × 7, 64, stride 2	7 × 7, 64, stride 2	$\left[3\times 3,\;64\;\right]\times 2$	55 × 55	11 × 11, 96, stride 4
2	56 × 56	3 × 3 max-pooling, stride 2	3 × 3 max-pooling, stride 2	3 × 3 max-pooling, stride 2	2 × 2 max-pooling, stride 2	27 × 27	3 × 3 max-pooling, stride 2
		$\left[\;\begin{array}{c} 1\times 1,\;64\\ 3\times 3,\;64\\ 1\times 1,\;256 \end{array}\;\right]\times 3$	$\left[\;\begin{array}{c} 1\times 1,\;64\\ 3\times 3,\;64\\ 1\times 1,\;256 \end{array}\;\right]\times 3$	$\left[\;\begin{array}{c} 1\times 1,\;64\\ 3\times 3,\;64\\ 1\times 1,\;256 \end{array}\;\right]\times 3$	$\left[\;3\times 3,\;128\;\right]\times 2$		5 × 5, 256, stride 1
3	28 × 28	$\left[\;\begin{array}{c} 1\times 1,\;128\\ 3\times 3,\;128\\ 1\times 1,\;512 \end{array}\;\right]\times 4$	$\left[\;\begin{array}{c} 1\times 1,\;128\\ 3\times 3,\;128\\ 1\times 1,\;512 \end{array}\;\right]\times 4$	$\left[\;\begin{array}{c} 1\times 1,\;128\\ 3\times 3,\;128\\ 1\times 1,\;512 \end{array}\;\right]\times 8$	2 × 2 max-pooling, stride 2	13 × 13	3 × 3 max-pooling, stride 2
					$\left[\;3\times 3,\;256\;\right]\times 3$		$\left[\;\begin{array}{c} 3\times 3,\;384\\ 3\times 3,\;384\\ 3\times 3,\;256 \end{array}\;\right]$, stride 1
4	14 × 14	$\left[\;\begin{array}{c} 1\times 1,\;256\\ 3\times 3,\;256\\ 1\times 1,\;1024 \end{array}\;\right]\times 6$	$\left[\;\begin{array}{c} 1\times 1,\;256\\ 3\times 3,\;256\\ 1\times 1,\;1024 \end{array}\;\right]\times 23$	$\left[\;\begin{array}{c} 1\times 1,\;256\\ 3\times 3,\;256\\ 1\times 1,\;1024 \end{array}\;\right]\times 36$	2 × 2 max-pooling, stride 2	1 × 1	3 × 3 max-pooling, stride 2
							average pooling
					$\left[\;3\times 3,\;512\;\right]\times 3$		[4096 fully connected, ReLU] ×2
							n-class fully connected, softmax
5	7 × 7	$\left[\;\begin{array}{c} 1\times 1,\;512\\ 3\times 3,\;512\\ 1\times 1,\;2048 \end{array}\;\right]\times 3$	$\left[\;\begin{array}{c} 1\times 1,\;512\\ 3\times 3,\;512\\ 1\times 1,\;2048 \end{array}\;\right]\times 3$	$\left[\;\begin{array}{c} 1\times 1,\;512\\ 3\times 3,\;512\\ 1\times 1,\;2048 \end{array}\;\right]\times 3$	$\left[\;3\times 3,\;512\;\right]\times 3$		
6	1 × 1	average-pooling			2 × 2 max-pooling, stride 2		
					[4096 fully connected, ReLU] ×2		
		n-class fully connected, softmax			n-class fully connected, softmax		

## Data Availability

The data presented in this study are available on request from the corresponding author.

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
