# Peer review of "Multi-View Convolutional Neural Networks in Rupture Risk Assessment of Small, Unruptured Intracranial Aneurysms"

_jpm, 2021, doi:10.3390/jpm11040239_

Round 1

Reviewer 1 Report

  1. What is the input image size for your network?
  2. It is not clear for me on how you train the whole structure. Did you train each view model independently first, then merge six models with view pooling and retrain the last few layers again? The training part is not clear.
  3. Lack of proper citations: Adam, ImageNet, label smoothing, AlexNet, VGG16, ResNet, Pytorch
  4. Figure 2 has a very poor resolution, and you should indicate epoch # in y-axis.
  5. The details of ROI cropping feature extraction are missing. How do you extract six different views of the same UIA?
  6. How did you calculate the confident interval (CI)?
  7. Did you use five-fold or 10-fold validation?
  8. Please also provide the F-1 score.

Author Response

I have attached the revision letter 

Reviewer 2 Report

The authors proposed a multi-view CNN assessing rupture risk in small UIAs. The proposal was tested on a relatively large dataset of UIAs (364 in training and 93 in test) and compared against some other networks. Experimental results show increased performance (sensitivity, specificity, and accuracy) when using a multi-view approach as opposed to a single view and top performance compared against the other networks.

While the work itself may be a contribution on its own, the way it has been presented is not adequate and up to the standards of the Journal of Personalised Medicine. My main concern is that overall, the paper is not easy to read. First, the main message (risk classification instead of diagnosis) seems lost in unnecessary discussion. This pattern is present since the start of the introduction of the work. Second, the lack of details throughout the entire manuscript impede assessing its quality. For example, essential details on dataset, image processing and considered networks is lacking and condensed elsewhere. The work needs to contain enough information so that readers can understand it clearly. Third, figures are of very low quality and explanations in captions are not enough. Fourth, results are extremely difficult to follow. While it is nice to have some numbers, and excess of them may be counterproductive. A more concise message would be helpful. Careful revision of the entire manuscript is necessary to cope with the afore issues.

Please, find some comments on the pdf attached.

Author Response

I have attached the revision letter as PDF form

Round 2

Reviewer 1 Report

  1. Although you add F1 score in Table 3, you do not mention it in any of your result section or abstract. Please also discuss F1 score in your results and abstract. You need to discuss F1 scores wherever other scores are mentioned.
  2. You should merge Dataset sections 2.1 and 3.1 and place the merged dataset section to section 2. You should introduce the dataset in the section of materials and methods.
  3. What are the contributions of this paper? Please explicitly write them down at the end of your Introduction section.
  4. You should have the epoch number at the x-axis of Figure 2.

Author Response

I have attached a revision letter. 

Reviewer 2 Report

The authors have addressed my comments successfully.

Author Response

No further reply